# Fluorescent Nanodiamonds for High-Resolution Thermometry in Biology

**DOI:** 10.3390/nano14151318

**Published:** 2024-08-05

**Authors:** Anna Ermakova

**Affiliations:** 1Physics Department, Hasselt University, Wetenschapspark 1, 3590 Diepenbeek, Belgium; anna.ermakova@uhasselt.be; 2Department of Magnetosphere-Ionosphere Coupling, Royal Belgian Institute for Space Aeronomy, 1180 Brussels, Belgium

**Keywords:** quantum sensors, fluorescent nanodiamonds, color center, nanoscale thermometry, temperature mapping, biological applications, living cells

## Abstract

Optically active color centers in diamond and nanodiamonds can be utilized as quantum sensors for measuring various physical parameters, particularly magnetic and electric fields, as well as temperature. Due to their small size and possible surface functionalization, fluorescent nanodiamonds are extremely attractive systems for biological and medical applications since they can be used for intracellular experiments. This review focuses on fluorescent nanodiamonds for thermometry with high sensitivity and a nanoscale spatial resolution for the investigation of living systems. The current state of the art, possible further development, and potential limitations of fluorescent nanodiamonds as thermometers will be discussed here.

## 1. Introduction

Temperature measurement within a single living cell is an important question for fundamental biological and medical sciences (for example, mitochondria activity, inflammations, etc.) as well as further industrial pharmacology applications (for instance, monitoring of photothermal cancer treatment) [1,2,3]. Due to this, various fluorescent markers attract significant attention and are applied to visualize temperature changes inside biological systems [4,5,6]. This list contains organic nanoparticles (e.g., fluorescent dye molecules, polymers, or proteins) [7,8], quantum dots [9], rare-earth doped nanoparticles [10], and nanodiamonds with color centers [11,12]. Fluorescent markers are an attractive system due to the low invasive influence of an optical measurement method. Such optical detectors are typically based on temperature-dependent fluorescent properties, such as spectral shift, changes in intensity, or lifetime. For biological applications, the stability and reliability of nanothermometers are crucial due to the initial complexity of investigated systems and the variation of their internal parameters (chemical components, pH level, viscosity, etc.) [13,14,15]. Therefore, the evaluation of suitable biological nanothermometers should be based not only on their sensitivity and parasitic influence but on environmental stability and reliability as well.

Organic fluorescent thermometers, including fluorescent dye molecules, fluorescent polymers, and fluorescent proteins, already recommended themselves very well for imaging and thermometry [7,8]. Depending on the particular type of nanoparticles, the temperature changes will provoke a shift of emission spectra or modify the fluorescent lifetime [4,6,7,8]. Their most significant advantages are ultra-small size (down to 1–2 nm for dye molecules) and various available functionalization protocols to control addressed attachment [16]. Organic nanoparticles cannot be used for long-term monitoring due to their photobleaching [17]. At the same time, talking about biological thermometry, their main problem is that their optical properties are not only temperature dependent but perform the response to the environment, for example, the pH level [18] or viscosity [8]. It makes temperature measurements less reliable and environmentally unstable. Therefore, it is hard to define whether the observed fluorescence changes were provoked by variations in temperature or other parameters.

Quantum dots are interesting optical marker systems. First, their fluorescent spectra can be efficiently designed by nanoparticle composition and size variation [19]. Such inorganic nanoparticles can also be functionalized for biological applications [19]. All these factors made them attractive systems for biological applications [19,20]. Temperature detection with quantum dots is performed by spectral shift observation or fluorescence intensity changes [6,9,21]. Nevertheless, they also have some disadvantages, particularly for biological applications. First, they are toxic materials [22]. Therefore, cell poisoning can cause temperature changes during the measurements, leading to parasitic influence and cell death. Second, their photo blinking, which makes them excellent marks for super-resolution imaging, prevents long-term monitoring [23]. Third, their fluorescence intensity, which is used for thermometry [21], also depends on the variation of the pH level [24].

Rare-earth doped nanoparticles, presented in various chemical compositions, are another type of inorganic nanoparticle that can be used for nanoscale thermometry [10]. Their temperature response can be observed in a few different ways, mainly changes in fluorescence intensity [10], the shift of emission spectra [10], or changes in lifetime [25]. The toxicity of rare-earth doped nanoparticles has not been entirely determined and remains under study [26]. Such nanoparticles are also used for sensing the chemical composition of the surrounding environment [27]. Since sensing protocols are based on the optical properties of rare-earth doped nanoparticles, environmental instability can also influence temperature measurements.

Another essential point for biological study, especially for investigating cellular metabolism, is the observation of not only temperature changes but also the measurement of the absolute temperature. The organic and inorganic fluorescent nanoparticles discussed above cannot provide this because their properties depend on environmental conditions. Thus, it is impossible to provide a one-to-one correspondence between fluorescence and temperature. It is highly critical for biological systems due to many simultaneous chemical reactions going on all the time [13,14,15]. For example, one of the most discussed papers about the absolute local temperature of mitochondria [28] claimed that it can reach up to 50 °C. However, such measurements were conducted with the fluorescent probe MitoThermo Yellow without correcting the parasitic environment-related fluorescence changes and are still under discussion in the community [29].

The alternative to the already-named sensors is fluorescent nanodiamonds (ND), whose optical properties are related to various types of lattice defects called color centers [11,12]. Optically active diamond defects are characterized by extreme photostability that allows long-term imaging and sensing [30]. One of the advantages of ND for biological study is that they are inert and nontoxic materials [31]. At the same time, the surface of ND can be chemically functionalized to make them more bioactive [32,33], for example, usable for drug delivery [34] or attachable to chosen intracellular parts [35]. Various diamond color centers offer different experimental protocols for temperature sensing (Figure 1). Currently, the three most commonly studied types of such optically active diamond defects are Nitrogen-Vacancy, Silicon-Vacancy, and Germanium-Vacancy. Each of them has its own advantages and disadvantages. Nevertheless, the most crucial reason to use ND for biological thermometry in opposition to previously named nanoparticles is the insensitivity of their fluorescent properties to the environment around diamond nanocrystals (pH level, ion concentrations, etc.), which will be described in detail later. Therefore, in the current review, NDs with these three types of well-known color centers will be deeply discussed, with a focus on their applications for thermometry for biological applications. Indeed, nothing is perfect, and diamond color centers are not, either; thus, additional stress will be given to possible limitations of ND utilization.

## 2. Fluorescent Nanodiamonds

The first temperature sensing protocols with diamond color centers were realized on bulk crystals. They are well and deeply summarized in the review [36]. The main optical properties of color centers are independent of the diamond size. Nevertheless, some important aspects, for example charge states, related to the diamond surface might influence the sensitivity of such sensors [37,38]. It becomes extremely valuable for biological thermometry, where nanodiamonds should be placed inside the living systems. This is why the following discussion will focus on fluorescent nanodiamonds selectively. The main parameters, such as type of ND, including color centers, size, fabrication method, and surface treatment, are presented in Table 1 and completed with the tested temperature range and achieved sensitivities.

### 2.1. Structure of Nanodiamonds

NDs have the same structure as a normal diamond lattice, with size starting from a few nm and increasing to large scales depending on final applications [58,59]. The fluorescent properties of ND are defined by optically active defects inside, known as color centers [11,12]. Such color centers can be created by impurity incorporation during diamond growth or afterwards by irradiation and annealing [58,60,61]. There are two main fabrication techniques: bottom-up and top-down.

Currently, the top-down method is the most popular for biological applications, due to the large amount of produced samples [59,62]. In this method, big diamond crystals are milled to the nanoscale size [58,63]. Such big diamond crystals are typically produced by the high-pressure high-temperature (HPHT) technique [64]. The chemical vapor deposition (CVD) diamond synthesis method can also be used. CVD provides high-quality diamond samples, but it is also more expensive [64]. Therefore, it is less common.

The bottom-up techniques include such types of samples as detonation ND [65], CVD ND growth on a substrate [66,67], and HPHT ND synthesized for a short time [68,69,70,71]. The detonation method delivers the smallest nanocrystals; however, it is more a combination of diamond cores with graphite shells [65,72]. CVD ND grown on a substrate cannot provide a large amount of the sample [73]. However, it can be a very powerful technique for fully controlled fabrication of individual ND or even microdiamonds for some applications [74]. The syntheses of HPHT ND for a short time can provide a large scale of nanocrystals with more uniform sizes and shapes in comparison to milled ones [68,69,70,71]. Thus, this technique has become more and more popular nowadays.

### 2.2. Thermometry with Nitrogen-Vacancy Centers

The most investigated and broadly used diamond color center is a Nitrogen-Vacancy (NV) defect consisting of a substitutional nitrogen atom next to a missed carbon—a vacancy. It can be found in two optically active charge states: neutral (NV^0^) and negative (NV^−^). At room temperature, their spectra demonstrate zero-phonon lines (ZPL) at 575 and 637 nm, respectively, with broad phonon sidebands [75]. Both optically active NV charge states can be used for temperature sensing. The detection method is based on observing the redshift of ZPL with a temperature increase [39,40,41,42]. Such a method is attractive due to the experimental simplicity since all operations and detection can be conducted purely optically. The ZPL shift happens linearly with temperature changes, simplifying temperature determination [39,40,41,42]. Many works focus on intracellular thermometry by detecting ZPL shift. As one example, the work of the group from Jinan University can be chosen (Figure 2a) [42]. Here, ND with NV centers were used for temperature sensing in different types of cells, particularly 4T1, C127, and HeLa cells. The temperature distribution was presented over a single C127 cell (Figure 2b). It demonstrates the temperature gradient and indicates that the nuclear close area is 5 °C warmer than the cell membrane region (Figure 2c). 

Another all-optical thermometry method with NV centers in ND was suggested in the paper [43]. In this work, only changes in the intensity of ZPL were observed for temperature detection. All measurements were taken for nanocrystals smaller than 50 nm. The experiments were performed with dry ND spin-coated on a quartz slide. To control the temperature of the system, the sample was placed in a heater with temperature stabilization within the accuracy of ±1 K. Such optical temperature observation was performed in the range of 295 K to 383 K [43]. The detected temperature dependence is not linear; however, there is a one-to-one correspondence. The achieved sensitivity was 300 mK Hz^−1/2^. Until now, such a measurement scheme has not been applied to real biological systems. The main disadvantage of such a technique for biological thermometry is that fluorescent intensity varies from one nanodiamond to another, which prevents reliable temperature mapping over the whole living cell.

For the NV^0^ defects, ZPL observation is the only possible method of thermometry. It is not the most efficient technique since only around 4% of NV fluorescence comes to ZPL [76]. Another measurement method is available for the NV^−^ center only. It is based on observing the optically detected magnetic resonance (ODMR) (Figure 1). NV^−^ defect has one extra electron from surrounding donors and, as a result, a spin equal to 1 with possible states m_s_ = 0, ±1. The spin-dependent fluorescence, explained by different relaxation processes for m_s_ = 0 and m_s_ = ±1, allows for the optical distinguishing of different spin states. The relaxation through the metastable state, which is more probable for m_s_ = ±1, is also responsible for spin conversion and polarization. Initially, ODMR, as well as spin relaxation times, were used for magnetic field detection, including the investigation of biological systems [77,78,79,80,81]. During such magnetometry experiments at different temperatures, it was observed by the group of D. Budker [82] that the positions of ODMR lines depend not only on the magnetic field due to the Zeeman effect (the split of ODMR) but also on temperature (the shift of the central ODMR frequency). The temperature increase provokes the shift of the central position of ODMR lines to the shorter microwave frequencies. Such an ODMR shift appears linearly with temperature, which makes ND a simple handle system for absolute temperature sensing. Soon after this first experiment, two groups simultaneously published papers about thermometry with NV^−^ centers in ND. The experiments provided in the Stuttgart group showed the sensitivity for ND-based sensors equal to 130 mK Hz^−1/2^ for a sample with an average nanocrystal size of 50 nm [44]. The Harvard team demonstrated the first case of intracellular thermometry by NV^−^ containing ND with a size of 200 nm and achieved a sensitivity of 9 mK Hz^−1/2^ [45]. The measured temperature increase was not biologically related but was artificially made and controlled by adding gold nanoparticles to convert light into heat. It was a proof-of-principle experiment and became the foundation of many following works [46,47,48].

Proof of the insensitivity to the environment of the thermometry ability of ND was demonstrated in the following work [47]. A group of scientists from Japan tested ND with NV^−^ centers for temperature sensing in different solvents, particularly in low-pH and high-pH solutions, high concentrations of Na^+^ and Cl^−^ ions, glycerol with high viscosity, surface polymer coating, and ethanol as an organic solvent (Figure 3a). For all those environments, the temperature dependence of the ODMR shift (D) was measured (Figure 3b). The results clearly show that there is almost no influence from the surrounding solution parameters (ion composition, viscosity, etc.) on the temperature sensing ability of NV containing ND. It allows us to conclude that NDs can be used inside cells as a robust thermometer without the need to correct the artificial negative influence of the environment.

Fluorescent ND can also be used for drug delivery and cancer treatment, as was demonstrated by many groups [34,35,83]. The extreme photostability of ND emission makes them attractive systems for theranostics applications, where long-term imaging and high sensing ability are combined with therapeutic effects. For instance, one such utilization of ND with NV^−^ centers was demonstrated by the group of T. Weil [84]. There, fluorescent NDs coated with nanogel shell (ND-NG) and additionally functionalized with indocyanine green molecules (ND-NG-ICG) were used to test cancer photo-treatment under infrared illumination with simultaneous intracellular thermometry (Figure 4a,b). Indocyanine green molecule is an approved imaging agent for different clinical applications [85]. Its ability to be used for cancer diagnosis and treatment is currently being investigated [85]. For such an application, the determination of optimal concentrations of molecules and light explosion for effective cancer phototherapy is essential. The ND-based thermometer system demonstrated in [84] allows us to evaluate the saturation temperature for different ND concentrations and, as a result, of indocyanine green in water (Figure 4c) and inside living cells (Figure 4d). The demonstrated measurements monitored the local temperature at the start of cell apoptosis, which is essential for improving phototherapy protocols.

Another possible biomedical application of ND-based thermometers is related to inflammation studies and other diseases. For example, neuron thermometry attracts significant attention from the scientific community due to brain inflammations [86] and different neurological diseases [87,88,89], as well as temperature-sensitive channels called transient receptor potential (TRP) channels [90], etc. Therefore, many groups worldwide have tried to apply fluorescent ND for neuron thermometry [49,50,52,91]. ND with NV centers were also proved to be applicable for thermometry in more complex living systems than just cells. For example, it was demonstrated that ND can be successfully used for measurements inside a living worm [50].

### 2.3. Thermometry with Silicon-Vacancy Centers

After the NV-based thermometry, Silicon-Vacancy centers (SiV) in diamond were discovered to be used for temperature sensing. SiV center consists of an interstitial silicon atom between two vacancies or a split vacancy. The single available measurement protocol for SiV-based nanoscale thermometers is that the whole observation is purely optical. Here, the thermometry is based on observing the redshift of ZPL (Figure 1) [52]. The general advantage of all-optical measurements in biology is related to the absence of an external microwave field, which is responsible for parasitic temperature increase. In contrast to NV centers, the ZPL of a SiV center is at 738 nm and dominates the spectrum, containing approximately 70% of all emitted photons [91]. It allows the collection of a higher number of photons for SiV than for NV during the same acquisition time. The near-infrared shifted fluorescence is also a significant advantage for investigating biological systems due to the near-infrared window in biological tissue [92]. All these make all-optical thermometry with SiV containing ND more efficient for biological experiments than for NV centers (Figure 1). In the case of ND, the first temperature-sensing experiment with SiV was performed for nanocrystals with a size of 200 nm and demonstrated a sensitivity of 521 mK Hz^−1/2^ [52]. For SiV defects, the redshift of ZPL with the temperature increase goes linearly, which allows the recalculation of absolute temperature changes from the SiV fluorescence spectrum. The first application of ND with SiV for intercellular thermometry was demonstrated in [53], where living HeLa cells were incubated with ND coated with human serum albumin (Figure 5b). There, the cell temperature was stabilized by an incubator to evaluate the sensing ability of ND with SiV in general (Figure 5c). Another advantage of ND with SiV centers is that its narrow near-infrared fluorescence can be separated from optical markers commonly used in biological study (Figure 5a), which makes dual imaging with simultaneous thermometry possible.

The broad sensing applications for biological systems with SiV centers in ND were explored further. The group of I. Vlasov presented a series of works about temperature probes with SiV defects to investigate living cells [54,55]. Here, the alternative method was applied. Instead of incorporating ND with SiV centers inside a living cell, ND with SiV was placed at the end of the glass pipette (Figure 6a) [55]. It allows the probe to be manipulated and controllably placed for temperature mapping (Figure 6b,c) [54]. This method was used to measure the temperature of the isolated mitochondria from the mouse brain [55]. These experiments demonstrated that the mitochondria temperature can experience a temperature difference of 4–22 °C and reach an absolute maximum of 45 °C.

### 2.4. Thermometry with Germanium-Vacancy Centers

Another diamond color center with high potential for thermometry is a Germanium-Vacancy center (GeV). It has a crystal structure similar to an SiV center. Its spectrum also predominantly consists of ZPL emission at 604 nm. Therefore, the experimental sensing protocol is based on observing the shift of ZPL (Figure 1), moving to the longer wavelengths with heating. The first demonstration of the GeV-based temperature sensor for a bulk diamond was made in the US-based research group [56]. However, ND with GeV are not yet as well used as nanocrystals with NV or SiV defects. In the last year, detonation ND containing GeV centers were recently tested for all-optical nanoscale thermometry [57]. There, the demonstrated sensitivity was at the level of 1 K Hz^−1/2^ (Figure 7). The obtained sensitivity is currently not as exciting as for NV or SiV centers. However, it is important to mention that, in general, detonation ND are significantly inferior in properties to CVD or HPHT nanocrystals, which were available for NV and SiV experiments earlier. Thus, these two types of nanodiamond synthesis attract a lot of attention and are actively used for the fabrication of GeV centers in ND nowadays [57,93,94]. Further improvement of the production of ND with GeV defects will increase the usability and sensitivity of diamond nanocrystals with that color center.

## 3. Discussion of Possible Limits

Fluorescent ND provide the outstanding possibility to measure temperature, especially intracellular temperature, independently from the environment [48], which is impossible for other currently used thermometers based on fluorescent nanoparticles [8,19,24]. The linear dependences of optical properties of various diamond color centers (ZPL or ODMR) allow for measuring absolute changes of variable temperature [42,44,45,52,56,57,82]. Detection of the absolute temperature of the system is fundamentally possible. However, there is an important point influencing the properties of fluorescent ND and absolute thermometry that must be addressed for the complete picture. NDs have a broad distribution in size, shape, and position of color centers inside. Any fabrication and subsequent separation of ND cannot provide a fully homologized sample. All that influences offsets for the central position of ODMR or intensity and broadening of ZPL that affect the detection of an absolute temperature. 

One of the first reports about such inhomogeneity of ND was presented in [95] regarding the initial splitting and shift of the ODMR line for NV centers in commercial ND. The samples were evaluated in a shielded environment to exclude external influence. Figure 8 shows the distributions of such ODMR shifts (a) and splits (b). It indicates that the absolute temperate determination from a single nanodiamond measurement can be problematic due to the natural variation of nanocrystal properties. A similar point was addressed in [48], where ND were tested in different solutions (Figure 3b). Minor variations of the temperature dependence of ODMR shift and, more importantly, its statistical distribution also demonstrated that absolute temperature can barely be found from a single spot measurement. However, it does not critically influence the observation of temperature changes monitored with a single nanocrystal. In such experiments, we are interested in the difference between two ODMR positions at two temperatures and its initial shift is canceling during calculations.

Similar inhomogeneity in the initial position and broadening of ZPL was observed for SiV centers in ND [53]. Here, the positions of ZPL were observed for temperatures ranging from 25 °C to 35 °C with a step of 2.5 °C, where the cell incubator stabilized the temperature of the solution (water). Two groups of ND with SiV can be selected: One demonstrates a good theoretically predicted shift of ZPL with the temperature increase. The second one has a low match with the theoretical calculation for SiV in a bulk diamond crystal. Therefore, additional investigation of color centers in ND according to their size and surface is essential for reliable absolute thermometry. A better understanding of nanodiamond materials will significantly improve the sensing protocols. It will influence thermometry with fluorescent nanodiamonds as well as magnetometry, which is also an attractive scientific direction.

## 4. Conclusions

Fluorescent NDs provide a unique opportunity for nanoscale thermometry with high sensitivity and nanoscale spatial resolution and are insensitive to the environment. The non-toxic properties of ND and different chemical functionalization methods allow for their application to biological study, where their sensing ability can be combined with theranostics applications. Nevertheless, NV centers recommended themselves as an outstanding system for biologically oriented magnetometry; effective thermometry based on them requires the application of an external microwave field that might bring limitations for intracellular use. Notably, the external microwave field can provoke a slight parasitic temperature increase, which can be crucial for biochemical reactions and cell biology. In the case of experiments with 3D tissue, deep measurements can be problematic since they will require high power of an external microwave field.

On the other hand, diamond color centers like SiV and GeV offer a fully optical measurement technique. Such an observation method is devoid of microwave-related problems and is highly promising for biological applications. At the same time, NDs with color centers provide the possibility of fast temperature mapping of living cells. The further development of such methods, together with the improvement of ND functionalization for addressed intracellular attachment, will provide a breakthrough method for cell biology and pharmacology, especially in combination with organ- and body-on-chip systems.

## Figures and Tables

**Figure 1 nanomaterials-14-01318-f001:**
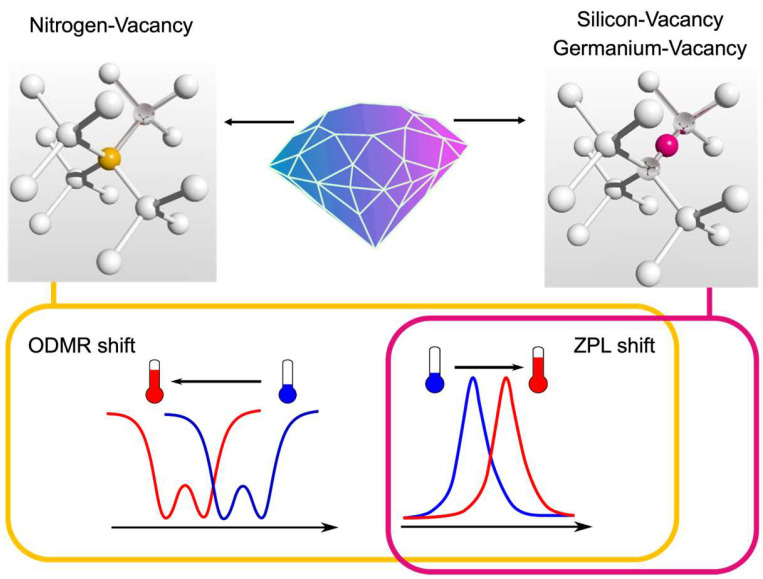
Crystal structures of diamond optically active defects: Nitrogen-, Silicon-, and Germanium-Vacancy centers. Temperature measurement protocols are based on the shift of optically detected magnetic resonance (ODMR) for the Nitrogen-Vacancy center only and on the shift of zero phonon line (ZPL) for all named defects.

**Figure 2 nanomaterials-14-01318-f002:**
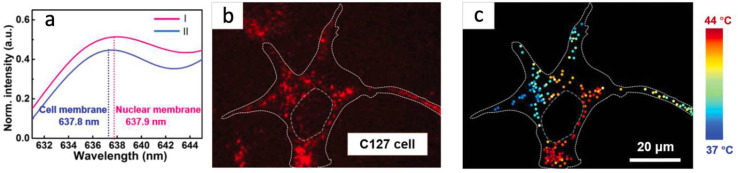
(**a**) Two spectra of aggregated ND, where line I represents the area close to the cell nuclear and line II—cell membrane. (**b**) Confocal image of fluorescent ND distributed inside a living C127 cell. (**c**) Temperature mapping recalculated from the ZPL shift for observed ND in (**b**). Data from [42].

**Figure 3 nanomaterials-14-01318-f003:**
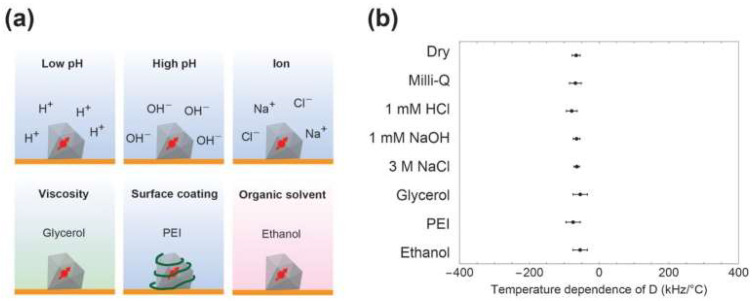
(**a**) ND in various types of solvents (low-pH, high-pH, high ion concentrations, high viscosity, surface polymer coating, organic solvent) to evaluate the influence of the environment on NV-based thermometry. (**b**) The temperature dependence of the ODMR shift (D) for different solvents. Data from [47].

**Figure 4 nanomaterials-14-01318-f004:**
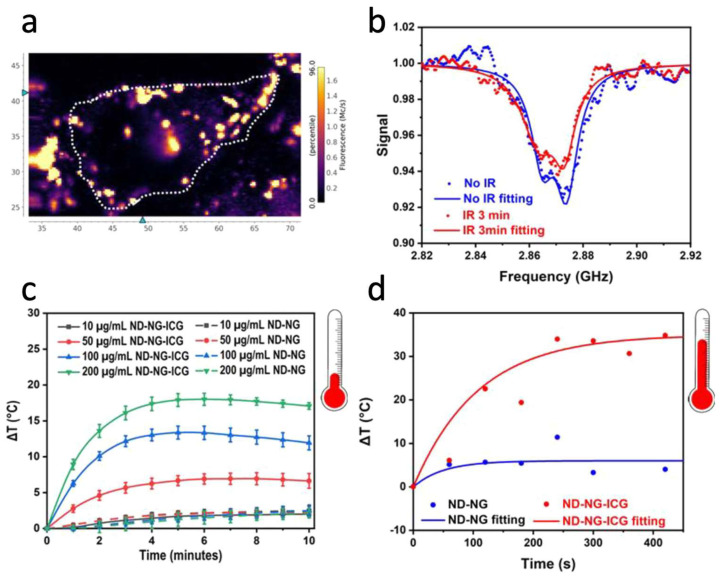
(**a**) Confocal image of ND coating with indocyanine green molecules in a living HeLa cell (the dashed line is the cell border) after 4 h of incubation. The concentration of coated ND is 10 μg/mL. (**b**) Selected ODMR spectra of coated ND with and without infrared irradiation (810 nm lamp; 0.35 W/cm^2^). (**c**) Thermal profiles of ND in water without (ND-NG) and with (ND-NG-ICG) indocyanine green with different concentrations under light irradiation (810 nm lamp; 0.35 W/cm^2^). (**d**) Intracellular thermometry and temperature saturation for NG-NG-ICG and ND-NG over 420 s under infrared irradiation (810 nm lamp; 0.35 W/cm^2^) for concentrations of 10 μg/mL. Data from [84].

**Figure 5 nanomaterials-14-01318-f005:**
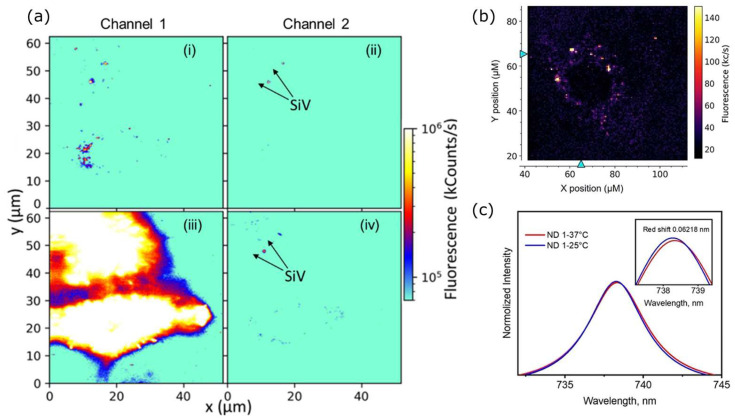
(**a**) Dual-confocal imaging of HeLa cells with ND with SiV: channel 1 collects all fluorescent >560 nm, and channel 2 is selective for SiV emission. (i) and (ii)—cells with ND only, (iii) and (iv)—cells additionally marked with membrane dye. (**b**) Confocal imaging of HeLa cells incubated with ND with SiV centers. (**c**) Thermometry by the redshift of SiV ZPL measured inside the living cell, where the incubator stabilized two temperatures. Data from [53].

**Figure 6 nanomaterials-14-01318-f006:**
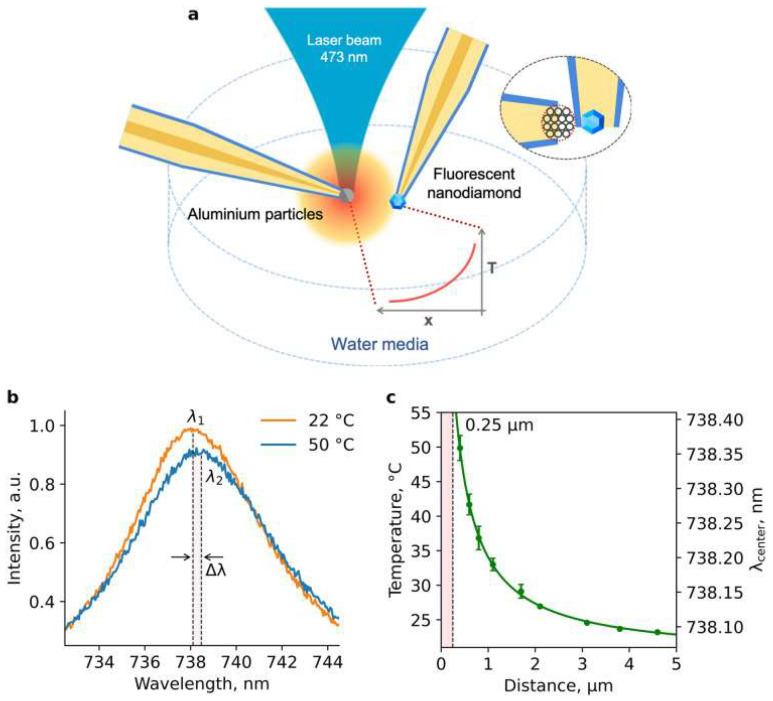
(**a**) Schematics representation of the thermometry probe based on SiV containing ND. (**b**) Photoluminescence spectra of the thermometer based on ZPL of SiV centers for two positions from the heater. (**c**) The shifts of SiV ZPL as a function of the distance and temperature from the heat source. Data from [54].

**Figure 7 nanomaterials-14-01318-f007:**
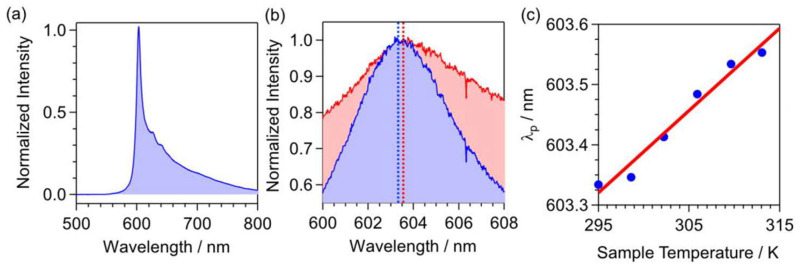
(**a**) Photoluminescence spectrum of GeV in detonation ND at a temperature of 22.0 °C. (**b**) The peak of the photoluminescence spectra of GeV centers was obtained at 22.0 °C (blue-filled) and 39.9 °C (red-filled). (**c**) The shift of the ZPL position depends on the temperature. Data from [57].

**Figure 8 nanomaterials-14-01318-f008:**
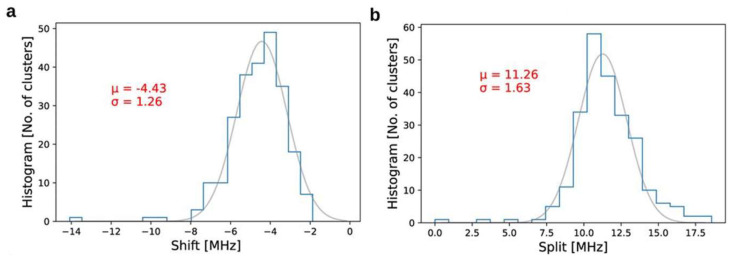
(**a**) The distribution of the ODMR shift for ND over 247 nanocrystals, where the axial strain is determined as ~4.43 MHz. (**b**) The distribution of the ODMR split for ND over 247 nanocrystals with the mean value of transverse strain at 11.26 MHz. Data from [95].

**Table 1 nanomaterials-14-01318-t001:** Summary of temperature measurements with different types of nanodiamond samples for biological or potential biological applications.

Color Center	Size, nm	Type	Measurement Method	Bio Tests	Temperature Range, °C	Sensitivity, K Hz^−1/2.^	Ref.
NV	100 ± 50	ND in PHEMA film	ZPL shift, Ensemble	No	35–120	0.15–1.1	[39]
NV	35	HPHT milled ND, carboxylated surface	ZPL shift	No	25–60	–	[40]
NV	70	HPHT milled ND, polyfunctional surface groups	ZPL shift	No	25–60	–	[40]
NV	100	HPHT milled ND, hydroxylated surface	ZPL shift	Yes	25–60	–	[40]
NV	100	Cationic surface, attached to gold nanorods	ZPL shift	Yes	28–75	2	[41]
NV	100 ± 30	Aggregated ND to microspheres	ZPL shift	Yes	20–60	0.5	[42]
NV	50	Milled ND	ZPL intensity	No	22–110	0.3	[43]
NV	50	Milled ND in polyvinyl alcohol	ODMR shift	No	22–40	0.13	[44]
NV	50	Milled ND	ODMR shift	Yes	20–30	0.009	[45]
NV	100	Milled ND, carboxylated surface	ODMR shift	Yes	33–36	1.5	[46]
NV	–	Milled ND	ODMR shift	Yes	27–37	–	[47]
NV	185	Milled ND	ODMR shift	Yes	28–38	3	[48]
NV	40 ± 15	HPHP milled ND in a cross-linked nanogel	ODMR shift	Yes	20–50	0.6	[35]
NV	20,000–30,000	Microdiamond on an optical fiber	ODMR shift	Yes	22–36	–	[49]
NV	100	Milled ND, carboxylated surface	ODMR shift	Yes	25–45	1.4	[50]
NV	11	Detonation ND	ODMR shift	No	22–42	0.36	[51]
SiV	200 ± 70	HPHP ND	ZPL shift	No	22–42	0.521	[52]
SiV	50	HPHT ND with HSA coating	ZPL shift	Yes	25–38	–	[53]
SiV	500	CVD ND on an optical fiber	ZPL shift	No	20–150	–	[54]
SiV	500	CVD ND on an optical fiber	ZPL shift	Yes	23–45	–	[55]
GeV	–	HPHT ND	ZPL shift	No	−125–125	0.3	[56]
GeV	30 ± 10	Detonation ND	ZPL shift	No	22–40	1	[57]

## Data Availability

Data are contained within the article.

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
