# Peer review of "Fluorescent Nanodiamonds for High-Resolution Thermometry in Biology"

_nanomaterials, 2024, doi:10.3390/nano14151318_

Round 1

Reviewer 1 Report (New Reviewer)

Comments and Suggestions for Authors

Summary

This paper is a review of the use of nanodiamonds for temperature sensing. It starts with a good summary of existing technologies (organic fluorophores, QDs, rare-earth doped nanoparticles) with their mechanisms and limitations. It introduces the colour centres and manufacturing methods. It goes on to describe methods and reported applications of ND thermometry including three different colour centres and compares sensitivity and practicality, including all-optical and ODMR-based methods. It is well organized and clearly written, divided into sections from a technology standpoint (rather than biological). It finishes with some practical manufacturing limitations (particle variability) and conclusions.

Completeness/Gap/Topic

The article addresses a promising area of quantum sensing, however, I do not see how (if at all) it improves on an existing review: Masazumi Fujiwara and Yutaka Shikano, Diamond quantum thermometry: from foundations to applications Nanotechnology, Volume 32, Number 48 (2021), which is very thorough and indeed addresses many of the limitations of this article that I highlight below. The existing review has more thorough methodological explanations, covers all the points the review covers, in more detail, with additional relevant concerns, such as surface preparation, selectivity and sensing artefacts, and more thorough descriptions of applications. This review is not even cited.

There are a number of crucial points that are missing that would make this review more useful (although I'm still unclear on how it would improve on Fujiwara et al.:

1. The most important omission is the biological context – sensitivities are quoted for various methods and listed in Table 1, but there is no discussion of how this relates to the biological need/pull. i.e. what are the most important biological questions that can be answered by thermometry? What magnitudes and timescales of these changes? How do these compare to reported sensitivities? What improvements are possible and needed?

2. Why is ODMR temperature sensing with NVs more sensitive than all-optical? Can this be explained wrt the Hamiltonian?

3. The author suggests that SiV all optical sensing is more sensitive due to more of the PL emission being in the ZPL, which makes sense. However, the sensitivity they report in line 255 (521 mK Hz-1/2) is lower than that reported for NVs (300 mK Hz-1/2 in line 158) – can the author explain this?

4. Table 1 is very instructive, demonstrating the large variation in the sensitivities reported in different publications. I would like to see a thorough, critical discussion in the text of the reasons for these differences, e.g. sensing protocols, optics, calculation. I would like the author to use these differences to highlight the key sensitivity considerations for ND thermometry, possibly with some methodological recommendations.

5. The authors state that ND thermometry is insensitive to solvents (pH, ions), however NV fluorescence intensity and ODMR peak size is sensitive to pH via T1 (T. Fujisaku, et al. pH Nanosensor Using Electronic Spins in Diamond. ACS Nano 13, 11726–11732 (2019)) and charge state (M. Sow et al. High-throughput nitrogen-vacancy center imaging for nanodiamond photophysical characterization and pH nanosensing. Nanoscale 12, 21821–21831 (2020)). Additionally, sources of magnetic noise such as reaction oxygen species will reduce ODMR contrast (V. Radu et al. Dynamic Quantum Sensing of Paramagnetic Species Using Nitrogen-Vacancy Centers in Diamond. doi: 10.1021/acssensors.9b01903 (2019)). Whilst it does seem to be true that the shift of the ODMR central frequency is independent of these factors, I think this should be clarified. Also relevant is line 160: if the ZPL shift is what is being measured (not intensity), why do intensity changes between NDs matter?

6. What is the spatial resolution, i.e. the sensing volume around a single ND? Temperature is an emergent thermodynamic quantity – what does it mean at such small scales? Are there any relevant statistical considerations?

Citations:

In the summary of manufacturing techniques – maybe also cite https://journals.aps.org/prapplied/abstract/10.1103/PhysRevApplied.20.044045 for CVD nanodiamonds.

Line 357: recommend citing https://arxiv.org/abs/2406.01181

I recommend the authors include the following reference for ND thermometry: https://pubs.acs.org/doi/full/10.1021/acsnano.3c05285

Comments on the Quality of English Language

Although the paper is well written and perfectly understandable, there are a few grammatical errors and idioms that sound slightly strange to a native speaker. It is also a bit wordy in places. I’ve listed a few below that affect the meaning:

Line 30-32: sentence meaning unclear.

Line 157: “leaner” should be “linear”.

Line 175: “more probable”

Author Response

Dear reviewer,

Thanks very much to the referee for the careful reading and for providing useful suggestions, which helped to improve the readability of the manuscript.

Comment 1:

The article addresses a promising area of quantum sensing, however, I do not see how (if at all) it improves on an existing review: Masazumi Fujiwara and Yutaka Shikano, Diamond quantum thermometry: from foundations to applications Nanotechnology, Volume 32, Number 48 (2021), which is very thorough and indeed addresses many of the limitations of this article that I highlight below. The existing review has more thorough methodological explanations, covers all the points the review covers, in more detail, with additional relevant concerns, such as surface preparation, selectivity and sensing artefacts, and more thorough descriptions of applications. This review is not even cited.

Response 1:

I would like to thank the referee for raising this question about the difference and the importance of this review. I agree with the referee that the review by Masazumi Fujiwara and Yutaka Shikano is comprehensive and provides a good explanation of diamond-based thermometry. It also touches on a variety of thermometric experiments. However, the submitted manuscript focuses in depth on the biological application of temperature sensing with nanodiamonds. The main idea of this work is to make it accessible to a broad community of biologists, therefore, it is not overloaded with information that does not represent an immediate importance for understanding diamond-based sensing. Moreover, the review by Masazumi Fujiwara and Yutaka Shikano does not sufficiently cover the experiments with nanodiamonds, which are the most important for intracellular thermometry. I agree with the referee that the reader who is interested in other aspects of diamond-based thermometry should be referred to the review by Masazumi Fujiwara and Yutaka Shikano, and I provided a citation.

Therefore, this submitted manuscript provides a review of the biological application of fluorescent nanodiamonds, including different fabrication techniques and various color centers, which were not included in the named review because this is a rapidly developing topic, and many new results were obtained after 2021.

Comment 2:

  1. The most important omission is the biological context – sensitivities are quoted for various methods and listed in Table 1, but there is no discussion of how this relates to the biological need/pull. i.e. what are the most important biological questions that can be answered by thermometry? What magnitudes and timescales of these changes? How do these compare to reported sensitivities? What improvements are possible and needed?

Response 2:

The list of important biological questions that can be answered by thermometry is quite long. It is shortly addressed by cited papers [old citations: 1-3, 25-26, 72-76]. For example, as explained through the text, it includes mitochondria thermometry, brain inflammation, and monitoring of photothermal cancer treatment. To clarify it more for the readers, as suggested, additional text and citations were added to the introduction.

Overall, biological systems are very complex for unambiguous definitions of the required temperature ranges and sensitivities. The thermometric effects in biology still remain poorly investigated. It leads to controversial statements in the literature. This is exactly the reason why nanothermometers represent high interest to the field and would help to shed light on the temperature-related effects and provide information about magnitude and timescale. The current review focuses on experimentally achieved parameters to attract more the attention of biologists to nanodiamond-based sensors and guide them through the large family of diamond color centers to target specific needs.

Comment 3:

  1. Why is ODMR temperature sensing with NVs more sensitive than all-optical? Can this be explained wrt the Hamiltonian?

Response 3:

I appreciate the referee's question on NV’s ODMR and optical sensitivities to the temperature. Both of these effects originate from the thermal expansion of the diamond lattice with the increase of temperature, but processes leading to the shift of the ODMR line and the shift of the ZPL are different. The ODMR measurements rest on the change of zero-field splitting parameter, which is caused by spin-spin interaction, and thus, it depends on the distance between these spins. The thermal expansion leads to the change of this distance. In the case of a zero-phonon line, the shift is caused by a change of electron-nucleus electrostatic interaction due to the displacement of nuclei with temperature. Precise calculation of the ODMR and ZPL shifts require density functional computation and cannot be found analytically. Therefore, in literature, temperature dependence is included in Hamiltonian only as parameters. For more information on sensitivity, I include two references: https://journals.aps.org/prb/pdf/10.1103/PhysRevB.90.235205 and https://journals.aps.org/prb/pdf/10.1103/PhysRevB.90.041201

Comment 4:

  1. The author suggests that SiV all optical sensing is more sensitive due to more of the PL emission being in the ZPL, which makes sense. However, the sensitivity they report in line 255 (521 mK Hz-1/2) is lower than that reported for NVs (300 mK Hz-1/2 in line 158) – can the author explain this?

Response 4:

I am not sure how this conclusion was drawn. Most probably, it was a misunderstanding. In the submitted paper there is no statement that “SiV all optical sensing is more sensitive”. It is written that “SiV containing ND more efficient than for NV”. The statement reflects the fact that ZPL of SiV center contains “70% of all emitted photons” allowing faster accumulation of the signal (the amount of detectable photons for SiV and NV centers are comparable). Also, the “near-infrared shifted fluorescence is a significant advantage for investigating biological systems due to the near-infrared transmission window in biological tissue. And the absence of MW field at 2.88 GHz does not contribute to the heating of water in cells. However, in the current context, “more efficient” is not equal to “more sensitive”. I thank the referee for raising this question since it helped to improve the text to prevent such misinterpreting. To highlight the connection between “more efficient” utilization of SiV all-optical thermometry for particular biological applications additional explanations are added (line 250-258).

Comment 5:

  1. Table 1 is very instructive, demonstrating the large variation in the sensitivities reported in different publications. I would like to see a thorough, critical discussion in the text of the reasons for these differences, e.g. sensing protocols, optics, calculation. I would like the author to use these differences to highlight the key sensitivity considerations for ND thermometry, possibly with some methodological recommendations.

Response 5:

I appreciate that the referee indicated this important point. Indeed, the sensitivity variation in different experiments, which use identical methodology, is large. I would like to underline that all the experiments can be divided by three methods. Within each method, theoretical sensitivity should be identically defined by the frequency shift per K and photon detection short noise (or instrumental noise). But in reality, sensitivity is rather defined by the far-from-ideal nanodiamonds. Even within one batch of the nanodiamonds, sensitivity distribution is huge. The reason is that biological experiments require large quantities of nanodiamonds, while the methods suitable for the fabrication of large sample amounts do not provide sufficient quality of the material. As a result, sensitivity is defined by the crystal strain, vicinity of the surface to the color center, presence of other defects, etc. There is a strong demand for nanodiamond quality improvement. However, this important topic still remains out of the scope of the researchers. To highlight this problem, additional text is added.

Comment 6:

  1. The authors state that ND thermometry is insensitive to solvents (pH, ions), however NV fluorescence intensity and ODMR peak size is sensitive to pH via T1 (T. Fujisaku, et al. pH Nanosensor Using Electronic Spins in Diamond. ACS Nano 13, 11726–11732 (2019)) and charge state (M. Sow et al. High-throughput nitrogen-vacancy center imaging for nanodiamond photophysical characterization and pH nanosensing. Nanoscale 12, 21821–21831 (2020)). Additionally, sources of magnetic noise such as reaction oxygen species will reduce ODMR contrast (V. Radu et al. Dynamic Quantum Sensing of Paramagnetic Species Using Nitrogen-Vacancy Centers in Diamond. doi: 10.1021/acssensors.9b01903 (2019)). Whilst it does seem to be true that the shift of the ODMR central frequency is independent of these factors, I think this should be clarified. Also relevant is line 160: if the ZPL shift is what is being measured (not intensity), why do intensity changes between NDs matter?

Response 6:

I completely agree with the reviewer that the surrounding solvents influence the intensity of fluorescence, the charge state of color centers, and the ODMR contract. And this is why the all-optical thermometry based on observation of NV intensity change (not ZPL shift) [55] (line 152) did not find applications in biology (line 160). To avoid misunderstanding in reading additional explanation is added (line 152). However, the temperature shift of the central position of ODMR is stable in different solvents, as demonstrated by work [66], especially in comparison to other optical markers [5, 15, 21, 24]

Comment 7:

  1. What is the spatial resolution, i.e. the sensing volume around a single ND? Temperature is an emergent thermodynamic quantity – what does it mean at such small scales? Are there any relevant statistical considerations?

Response 7:

The spatial resolution depends on the surrounding medium and its heat transfer properties. Therefore, nanoscale sensors do not really measure nanoscale environments but rather the volume within which equilibrium is reached faster than the measurement time. Thus, there is no contradiction to the temperature definition as an average kinetic energy of microscopic volume. Although it is quite simple to simulate the heat transfer and temperature distribution for the homogeneous media, this is not the case for living cells. Living cells have complex structures with various components and additional local heat sources such as mitochondria https://rupress.org/jgp/article-pdf/152/8/e202012629/1807655/jgp_202012629.pdf . So far there were no satisfactory simulations of temperature distribution within a cell. In this case, experimental data would be useful to compare them with simulations and improve models. Mapping the temperature throughout the cell with a large number of measurement points on a certain timescale would be extremely useful to adjust the simulations. Unfortunately, we are still at the stage of data collection.

Comment 8:

Line 30-32: sentence meaning unclear.

Response 8:

Corrected

Comment 9:

Line 157: “leaner” should be “linear”.

Response 9:

Corrected

Comment 10:

Line 175: “more probable”

Response 10:

Corrected

Best regards,

Anna

Reviewer 2 Report (New Reviewer)

Comments and Suggestions for Authors

In this review paper, the author focuses on color centers in diamond for applications in temperature measurements in biology. Such this review paper is very  needed and useful, allowing for a quick assessment of the current state of knowledge in a given field.

The author focuses here on the three most popular crystal structure defects of diamond, i.e., NV, SiV, and GeV. The article reads well and is suitable for publication after minor corrections.

line 338 - it is necessary to standardize the writing of the temperature unit after or without a space throughout the text:

"25°C to 35°C with a step of 2.5 °C," 

line 483 - title written without spaces

line 498 - missing year

 I don't see any new articles from 2024 like: 

https://doi.org/10.1063/5.0201154

https://doi.org/10.1364/BOE.524293

etc.

Author Response

Dear reviewer,

Thank you very much for your positive evaluation of the manuscript for publication, careful reading, and useful suggestions regarding typos. I appreciate your opinion about the usefulness and necessity of this review. Thank you for the precise check even of the citation. All mentioned mistakes are corrected.

Comment 1:

line 338 - it is necessary to standardize the writing of the temperature unit after or without a space throughout the text: "25°C to 35°C with a step of 2.5 °C,"

Response 1:

Corrected through the text.

Comment 2:

line 483 - title written without spaces

Response 2:

Corrected.

Comment 3:

line 498 - missing year

Response 3:

Corrected.

Comment 4:

I don't see any new articles from 2024 like: 

https://doi.org/10.1063/5.0201154

https://doi.org/10.1364/BOE.524293

Response 4:

I agree that a few recent papers are missing. This is corrected, and relevant information is added to the table.

Best regards,

Anna

Reviewer 3 Report (New Reviewer)

Comments and Suggestions for Authors

The presented paper aims to compile, evaluate, and critically discuss the extensive work and publications on temperature measurement at the micrometer scale using nanoparticles. In my opinion, the author achieves this in an excellent manner. The paper primarily focuses on thermometry in biology. Initially, it addresses various nanoparticles used for such purposes, describing their capabilities and limitations.

Most of the review is then dedicated to nanodiamonds. The author first discusses the advantages and disadvantages of nanodiamonds compared to other nanoparticles in a very critical and comprehensive manner. The paper examines nanodiamonds based on their different production methods and discusses the underlying mechanisms of fluorescence-based thermometry. Fluorescence-based thermometry is discussed in terms of the various possible color centers, and the advantages and disadvantages of the two measurement methods and the three different color centers are extensively covered.

The author succeeds exceptionally well in selecting illustrative demonstration examples for all cases and cites well-chosen examples from the literature. The concluding discussion focuses mainly on the negative influence of the inhomogeneities of nanodiamonds on the thermometry methods discussed. This issue is also extensively and critically discussed. The summary appears concise, precise, and well-chosen considering the paper.

Overall, this is a very readable review that excellently research, evaluates, and summarizes a very current topic. I enjoyed reading this paper.

However, I would like to make one suggestion. The paper is titled: "Fluorescent Nanodiamonds for Nanoscale Thermometry in Biology." From the discussion, the measurements for thermometry are spatially limited by the Abbe limit. Thus, each nanodiamond can only be detected with an accuracy of about 150-250 nm. The cited works use nanodiamonds that are mostly larger than 50 nanometers. Therefore, I would not use "nanoscale thermometry" in the title. Despite the use of nanoparticles, the spatial resolution is more likely in the micrometer range and at best in the range above 200 nm.

Congratulations on this excellent work!

Comments on the Quality of English Language

No additional comments.

Author Response

Dear reviewer,

Thank you very much for the positive evaluation of this manuscript for publication. I am happy to see that you found this review well-written, structured, critical enough, and enjoyable to read.

Comment 1:

However, I would like to make one suggestion. The paper is titled: "Fluorescent Nanodiamonds for Nanoscale Thermometry in Biology." From the discussion, the measurements for thermometry are spatially limited by the Abbe limit. Thus, each nanodiamond can only be detected with an accuracy of about 150-250 nm. The cited works use nanodiamonds that are mostly larger than 50 nanometers. Therefore, I would not use "nanoscale thermometry" in the title. Despite the use of nanoparticles, the spatial resolution is more likely in the micrometer range and at best in the range above 200 nm.

Response 1:

I am grateful to the referee for this note. Indeed, in most of the cases, the measurements address much large volume. It is not only about the resolution of the confocal microscope (where super-resolution imaging can be applied) that gives a few hundred nanometers (or a few tens of nanometers for super-resolution) but also the thermalized volume around the nanodiamond. Although, there are a few points why “nanoscale” can still be used in the title, I agree that “high-resolution thermometry” or “sub-micron thermometry” would be a more appropriate title to avoid confusion. According to this, the title is changed to “Fluorescent nanodiamonds for high-resolution thermometry in biology”.

Best regards,

Anna

This manuscript is a resubmission of an earlier submission. The following is a list of the peer review reports and author responses from that submission.

Round 1

Reviewer 1 Report

Comments and Suggestions for Authors

The manuscript titled “Fluorescent nanodiamonds for nanoscale thermometry in biology” by Ermakova, A. is a Review work where the author shows the most recent advances in this field highlighting the potential applications in biological systems with some relevant recently reported examples. The manuscript is generally well-written and this is a topic of growing interest.

However, it exists some points that need to be addressed (please, see them below detailed point-by-point) to improve the scientifc quality of the submitted manuscript paper before this article will be consider for its publication in Nanomaterials.

1) KEYWORDS. The author should consider to add the term “quantum sensors” in the keyword list.

2) INTRODUCTION. The author clearly depicts the state-of-the-art of this examined field. In some points further information details should be provided. For example, “Quantum dots are interesting optical marker systems (…) also depends on the variation of the pH level” (lines 34-39). Here, it may be convenient to show the possibility in the employment of quantum dots to produce fluorescent self-healing assemblies for biomedical applications [1].

[1] Liu, C.; et al. Quantum Dots-Loaded Self-Healing Gels for Versatile Fluorescent Assembly. Nanomaterials 2022, 12, 452. https://doi.org/10.3390/nano12030452.

3) FLUORESCENT NANODIAMONDS. “The alternative to already-name sensors is fluorescent nanodiamonds (ND) (…) will be deeply discussed” (lines 52-63). Here, the authors shows the main advantages displayed by the fluorescent nanodiamonds but it may be desirable also to highlight the existing limitations in comparison to other alternative approaches.

4) Figure 2, panel a (line 84). The lettering is slightly blurry. Maybe this issue comes from the PDF converstion step. The authors should contact the assistant Editor in order to fix this problem. Same comment for the Fig. 3 panel a (line 123) and Fig. 4 panels b-d (line 140).

5) “Another measurement method is available of the NV center (…) the spin-dependent fluorescence, explained by different relaxation processes (…) were used for magnetic field detection, including the investigation of biological systems (…) central ODMR frequency” (lines 91-102). Here, even if I agree with this statement provided by the authors it should not be neglected the potential combination of the data gathered by optical magnetic resonance by other single-molecule techniques [2] to sense with ultrahigh sensitivity the local magnetic response of magnetic nanomaterials.

[2] Winkler, R.; et al. A Review of the Current State of Magnetic Force Microscopy to Unravel the Magnetic Properties of Nanomaterials Applied in Biological Systems and Future Directions for Quantum Technologies. Nanomaterials 2023, 13, 2585. https://doi.org/10.3390/nano13182585.

6) “2.1. Thermometry with Silicon-Vacancy centers” (lines 156-195). Here, in addition to the promising biological applications found for this technology it should not be neglected the relevance in the development of quantum sensors [3].

[3] Segawa, T.F.; et al. Nanoscale quantum sensing with Nitrogen-Vacancy centers in nanodiamonds – A magnetic resonance perspective. Prog. Nucl. Magn. Reson. Spectrosc. 2023, 134-135, 20.38. https://doi.org/10.1016/j.pnmrs.2022.12.001.

7) “2.1 Thermometry with Germanium-Vacancy centers” (lines 196-214)”. First, the authors should relist this subsection as “2.2.”. Then, this section should be expanded for other metal vacancies in metal oxides as titanium dioxide (TiO2) or zinc oxide (ZnO) which can treasure enormous applications in gas sensing, photocatalysis and the development of efficient solar cells, among others.

8) DISCUSSION AND POSSIBLE LIMITS. The information furnished in this section is accurate. No actions are requested for the author.

9) CONCLUSIONS. This section perfectly remarks the most relevant outcomes found by the authors in this field. The author should add a brief statement to discuss about the future line actions to pursue this research and the open perspectives.

Comments on the Quality of English Language

The manuscript is generally well-written albeit it may be desirable if the author could recheck the manuscript in order to polish final details susceptible to be improved. 

Author Response

Dear reviewer,

Thank you so much for your careful reading and suggestions.

  • Thank you for the suggestion. It will be added.
  • An overview of the main properties of quantum dots and organic nanoparticles for temperature sensing is given for comparison. The possible applications of such systems are out of the focus of the current paper review.
  • The limitations of fluorescent nanodiamonds are deeply discussed in part 3, “Discussion of possible limits.” In my view, it makes little sense to talk about them in the state-of-the-art part before their sensing ability is described.
  • -
  • Thank you for the suggestion; it is an interesting review. However, I have to admit that the current review does not focus on magnetic sensing. It mentions magnetometry by NV centers in nanodiamonds just to save the historical logic of NV study and applications.
  • The suggested paper is a great review of NV magnetometry. The small part related to temperature measurement in this paper is already discussed in the current review, which cites the original papers, which were pioneering work in NV thermometry.
  • Thank you for the suggestion. However, the current review discusses only fluorescent nanodiamonds as optical markers for biology and does not touch any other thermometry systems.

Best regards,

Anna Ermakova

Reviewer 2 Report

Comments and Suggestions for Authors

This manuscript is a brief overview introducing the problem of using nano-sized diamonds for thermometry for biological and medical applications. The advantages of the review are its brevity and detailed discussion of the results of modern publications on diamond nanoscale thermometry. The main drawback is the absence in the text of the manuscript original proposals and ideas for the development of nanodiamond thermometry technology.

Some of the statements in the manuscript are not sufficiently substantiated and are not supported by references to publications. In my opinion, not enough attention is paid to alternative technologies and other fluorescent thermometers. This can be corrected by pointing to the data given in the article [Đačanin Far, L., & Dramićanin, M. D. (2023). Luminescence Thermometry with Nanoparticles: A Review. Nanomaterials, 13(21), 2904].

The author of the review discusses only methods based on the magnitude of the line shift depending on temperature, however it is also possible to measure local temperature by changes in ZPL height by referencing its phonon sideband using a ratiometric technique developed in [Plakhotnik, T., Aman, H., & Chang, H. C. (2015). All-optical single-nanoparticle ratiometric thermometry with a noise floor of 0.3 K Hz− 1/2. Nanotechnology, 26(24), 245501].

In my opinion, it would be desirable in a review to discuss in more detail the effect of nanodiamond surface functionalization on the sensitivity of fluorescent thermometers to external environmental conditions and indicate the most optimal functionalization parameters for measurements in various biological media.

The review lacks tables that include an overview of the most recent literature reports, which would indicate the parameters of nano-sized diamonds, the temperature range, objects and measurement methods, as well as the achieved sensitivity values. Such a table (or tables) may be published as a Supplement.

After making additions to the text of the manuscript, it can be published in the journal Nanomaterials.

Author Response

Dear reviewer,

Thank you so much for your careful reading and suggestions.

Other fluorescent thermometers are out of the current paper. Some of them, which are relevant for biological sensing, are briefly discussed for comparison. However, the deep discussion is not related to the current topic. There are many interesting and complete reviews about different types of optical thermometers. Some links are provided. It is not possible to add all of them due to their high number.

Thanks for the great recommendation. This measurement technique has not yet been applied to biological systems. Nevertheless, I agree that it is valid information to discuss within the current review. It is now added to the paper.

I completely agree that it would be great to include the discussion of surface possible influence. However, such research has not yet been performed for thermometry with nanodiamonds. I am also very interested in seeing such results.

Best regards,

Anna Ermakova

Round 2

Reviewer 1 Report

Comments and Suggestions for Authors

After careful assessment, I agree with the comments raised by other reviewers indicating the content of this manuscript is not enough to be considered as a Review work. For it, it is recommended a deep restructuring process providing many more relevant insights for the examined field. 

Author Response

Dear reviewer,

Thank you for your review report.

I would like to mention, first of all, that the currently submitted review has a very particular peruse, discussing fluorescent nanodiamonds as thermal sensors for biological applications.

There are several reviews about thermally active nanoparticles. However, nanodiamonds are still quite a new system for such applications and they are not well presented there. In the case of published reviews focused on nanodiamonds, in general, they have a main stress on the magnetometry with NV centers since this topic is well more established. The thermometry part is usually mentioned briefly at the end and typically for NV centers only.

Here, on the opposite, different diamond color centers are discussed as thermal detectors. The advantages and disadvantages of each of them were discussed separately. That allows us to get an overview of fluorescent nanodiamonds for biological thermometry and guides further research activity.

I agree completely with one of the reviewers that it would be great to discuss the surface influence of thermal sensing of nanodiamonds. Unfortunately, such works have not yet been performed, and no systematic study of size influence has been made as well. Such aspects are mentioned in the discussion part of the submitted paper.

I think that it is important to keep the focus of the review on one particular topic (as thermometry with fluorescent nanodiamonds in that case), instead of combining previously published reviews together. Therefore, I see that the review is complete according to the achieved results in the community of fluorescent nanodiamonds for biological thermometry.

Best regards,

Anna Ermakova